# Targeting Transcriptional CDKs 7, 8, and 9 with Anilinopyrimidine Derivatives as Anticancer Agents: Design, Synthesis, Biological Evaluation and In Silico Studies

**DOI:** 10.3390/molecules28114271

**Published:** 2023-05-23

**Authors:** Razan Eskandrani, Lamees S. Al-Rasheed, Siddique Akber Ansari, Ahmed H. Bakheit, Abdulrahman A. Almehizia, Maha Almutairi, Hamad M. Alkahtani

**Affiliations:** 1Department of Pharmaceutical Chemistry, College of Pharmacy, King Saud University, P.O. Box 2457, Riyadh 11451, Saudi Arabia442202895@student.ksu.edu.sa (L.S.A.-R.); sansari@ksu.edu.sa (S.A.A.);; 2Drug Exploration and Development (DEDC), Department of Pharmaceutical Chemistry, College of Pharmacy, King Saud University, Riyadh 11451, Saudi Arabia

**Keywords:** 2-anilinopyrimidine, CDK, antiproliferative activity, molecular docking

## Abstract

Cyclin-dependent kinases (CDKs) are promising targets in chemotherapy. In this study, we report a series of 2-anilinopyrimidine derivatives with CDK inhibitory activity. Twenty-one compounds were synthesized and their CDK inhibitory and cytotoxic activities were evaluated. The representative compounds demonstrate potent antiproliferative activities toward different solid cancer cell lines and provide a promising strategy for the treatment of malignant tumors. Compound **5f** was the most potent CDK7 inhibitor (*IC*_50_ = 0.479 µM), compound **5d** was the most potent CDK8 inhibitor (*IC*_50_ = 0.716 µM), and compound **5b** was the most potent CDK9 inhibitor (*IC*_50_ = 0.059 µM). All the compounds satisfied the Lipinski’s rule of five (molecular weight < 500 Da, number of hydrogen bond acceptors <10, and octanol–water partition coefficient and hydrogen bond donor values below 5). Compound **5j** is a good candidate for lead optimization because it has a non-hydrogen atom (N) of 23, an acceptable ligand efficiency value of 0.38673, and an acceptable ligand lipophilic efficiency value of 5.5526. The synthesized anilinopyrimidine derivatives have potential as anticancer agents.

## 1. Introduction

Cancer is the second leading cause of death in the world [1,2]. It is a result of uncontrolled cell proliferation and differentiation, followed by metastasis, as a consequence of inherited genes or exposure of the body to chemicals, radioactivity, or infectious agents [3,4]. The understanding of the molecular mechanisms underlying carcinogenesis has helped in the development of novel anticancer agents that cause minimal damage to normal cells [5]. Douglas Hanahan and Robert A. Weinberg described 10 hallmarks associated with cancer cells that have led to the development of new anticancer agents [6]. These hallmarks include self-sufficient growth signals, limitless replication potential, sustained angiogenesis, insensitivity to anti-growth signals, evasion of apoptosis, tissue invasion and metastasis, genome instability and mutation, inflammation, avoidance of immune destruction, and abnormal metabolism [7].

Cyclin-dependent kinases (CDKs) are a family of serine/threonine protein kinases that contain catalytic kinase and regulatory cyclin subunits. The CDK family is classified into the following two groups according to the main function: the cell-cycle-related group (CDK1, 4, and 5) and the transcriptional group (CDK7, 8, 9, 11, and 20) [8,9]. Abnormal function or high levels of CDKs have been observed in many types of cancers, particularly hematological malignancies, which contribute to at least one hallmark of cancer [6,10]. Several small molecules were designed and synthesized to target CDKs (either transcriptional or cell cycle CDKs). Many of these molecules, such as palbociclib, ribociclib, and abemaciclib, were approved by the Food and Drug Administration (FDA) as CDK inhibitors for the treatment of hematological and solid malignancies (Figure 1) [11,12].

CDK9 is a member of the CDK family, and its partner cyclin T forms the positive transcription elongation factor b (p-TEFb) complex [13]. This complex plays a vital role in transcriptional regulation via phosphorylation of the RNA polymerase II C-terminal domain at Ser24 [14]. CDK9 plays a crucial role in regulating the transcription of several genes involved in cell growth, cell cycle progression, and apoptosis, such as Myc, a proto-oncogene, and Mcl-1, an antiapoptotic member of the Bcl-2 family [15]. Therefore, CDK9 inhibition reduces transcription and inhibits the expression of target genes that control the proliferation and survival of cancer cells.

Recently, some small surrogates targeting CDK9 were reported, including selective and pan-CDK inhibitors [14]. Pan-CDK inhibitors, which inhibit CDK9, in addition to other CDK isoforms, were the first generation CDK inhibitors evaluated in clinical trials. Flavopiridol was the first CDK9 inhibitor to enter clinical trials for the treatment of chronic lymphocytic leukemia (Figure 2) [16]. However, its further development was constrained by its poor selectivity and side effects. Second-generation inhibitors, which are selective CDK9 inhibitors, including atuveciclib [17], BAY-1251152 [18], LS-007 [19], and ly-2857785 [20], were subsequently discovered (Figure 2).

By studying the structures of previously reported kinase inhibitors, we found that the vast majority of these inhibitors possess pyrimidine as either 2-anilinopyridine or 2-anilinopyrimide as a pharmacophore in their structure [21,22,23,24,25,26,27]. In this study, we aimed to synthesize pyrimidine derivatives, either 2-aminopyrimidine or 2-anilinopyrimidine, with a different hydrophobic system (phenyl, pyridinyl, or pyrimidinyl) at position 4 of the pyrimidine ring, attached to either electron-withdrawing or electron-donating groups, which could affect the activity of the target compounds. In addition, position 5 of the pyrimidine ring was designed to be occupied either by a proton or cyano group that was reported to be in contact with the gatekeeper residues, allowing the study of the structure–activity relationship of these compounds as effective antitumor candidates with potential CDK inhibitory activities (Figure 2).

## 2. Results

### 2.1. Chemistry

The strategy employed to synthesize 2-anilinopyrimidine derivatives is outlined in (Figure 1). The synthesis began with the preparation of eneaminones (**2a**–**m**) by heating commercially available substituted acetophenones (**1a**–**m**) and *N*,*N*-dimethylformamide dimethyl acetal (DMF-DMA) in toluene (PhMe) or *N*,*N*-dimethylformamide (DMF) [28,29]. Subsequently, cyanimide was used as the starting material for the preparation of guanidine (**4a**,**b**), as reported in the literature [30]. Then, 3-Amino-benzene/sulfonamide was refluxed with cyanamide and chlorotrimethylsilane (TMSCl) in acetonitrile to obtain the desired 3-guanidino-benzene/sulfonamide (**4a**,**b**). The final step was the reaction of enaminones with guanidine nitrate or guanidino-benzene/sulfonamide in *n*-butanol in the presence of K_2_CO_3_ to obtain the target compounds (**3a**–**h**) and (**5a**–**m**), respectively [31].

The newly synthesized anilinopyrimidines **5a**–**m** were characterized by their melting point and spectroscopic data (^1^H-NMR, ^13^C-NMR, and MS). The ^1^H-NMR spectra of all synthesized target compounds **5a**–**m** showed a characteristic singlet signal at ~10.1 ppm, corresponding to the amino proton -NH-, confirming the formation of our target derivatives. In addition, the singlet signal with two equivalent protons at ~7.3 (-SO_2_NH_2_ group) revealed the presence of a sulfonamide group in the final targets **5a**–**l**. Furthermore, the IR spectra of anilinopyrimidines **5a**–**m** exhibited characteristic absorption bands at ~3300 cm^–1^ owing to the presence of amino and sulfonamide groups. In addition, the cyano derivative compounds **5k**–**m** exhibited strong absorption at ~2200 cm^–1^ owing to the presence of the CN group.

^13^C-NMR spectra revealed the presence of at least three peaks above 159 ppm in **5a**–**m**, indicating the formation of pyrimidine (related to carbons adjacent to N atoms). All other spectral and analytical data were consistent with the assumed structures. The mass spectra of the final targets displayed the correct molecular ion peaks (M^+^), as suggested by their molecular formulae.

### 2.2. In Vitro Antiproliferative Activities

CDK9 is reported to have a role in the proliferation and progression of several solid tumors such as colorectal, hepatic, cervical, and breast cancers [32,33,34,35,36,37]. Therefore, the in vitro antiproliferative activities of the newly synthesized 2-aminopyrimidines and 2-anilinopyrimidines were evaluated using the standard MTT assay [38,39] against four human cancer cell lines, namely, HCT116 (colorectal carcinoma), HepG2 (hepatocellular carcinoma), HeLa (cervical epithelioid carcinoma), and MCF7 (mammary gland breast cancer), using doxorubicin (DOX) as a positive control. The cell lines mentioned above were used to determine the inhibitory effects of the compounds on cell growth using the MTT assay. The *IC*_50_ (µM) values showing the in vitro cytotoxicity of the tested compounds are summarized in Table 1. The tested compounds exhibited variable degrees of anticancer activity against the selected cancer cell lines.

Compounds **3b**, **3f**, **5a**, and **5d** displayed remarkably strong anticancer potencies against HepG2 cells, with *IC*_50_ of 18.85, 8.18, 7.84, and 11.72 µM, respectively. Compounds **5c** and **5h** showed moderate activity with *IC*_50_ of 24.39 and 28.03 µM, respectively, compared to the positive control, DOX (*IC*_50_ = 4.50 µM). 

Compound **5m** had the highest anticancer activity against HeLa cells, with an *IC*_50_ of 9.83 µM. Compounds **3b**, **3f**, and **5a** showed higher inhibitory activity, with *IC*_50_ values of 15.18, 18.24, and 11.51 µM, respectively. Furthermore, compounds **5c**, **5d**, and **5h** had moderate anticancer activity, with *IC*_50_ of 29.76, 22.21, and 25.38, respectively, compared to the positive control DOX (*IC*_50_ = 5.57 µM).

Comparison of *IC*_50_ values against MCF-7 cells revealed that compounds **3b**, **3f**, **5a**, **5c**, **5d**, and **5h** showed significant anticancer potency with *IC*_50_ of 10.21, 6.08, 3.84, 16.13, 8.72, and 14.29 μM, respectively. Compound **5a**, with *IC*_50_ = 3.84 μM, was the most active among the tested compounds with anticancer activity against MCF-7 cells higher than that of the positive control DOX (*IC*_50_ = 4.17 μM). Compounds **3a** and **3e** displayed moderate anticancer potency with *IC*_50_ of 21.81 and 26.30, respectively. 

All the tested compounds displayed different antitumor activities, ranging from weak to moderate, against HCT116 cells.

Among the 2-aminopyrimidine analogs bearing H at position 5, compound **3b**, with a small electron withdrawing substituent (Cl) on the phenyl group at position 4 of the pyrimidine ring, was the most potent compound with *IC*_50_ of 18.85, 15.18, and 10.21 µM against HePG-2, HeLa, and MCF-7, respectively. On the contrary, compounds with the substituents OCH_3_, CF_3_, and OCF_3_, as in compounds **3c**, **3d**, and **3e**, showed weak activity (*IC*_50_ ranging from 39.13 to >100 µM). Furthermore, compounds without substituents, such as **3a** (*IC*_50_ ranging from 21.18 to 37.29 µM), showed a weak potency. In addition, the 2-aminopyrimidine derivative **3f**, with unsubstituted phenyl at position 4 and CN at position 5, exhibited potent activity against HepG-2, HeLa, and MCF7 cells, with *IC*_50_ of 8.18, 18.24, and 6.08 µM, respectively, compared with that of its analogs with a substituted phenyl at position 4 (compounds **3f**–**h)**.

The 2-anilinopyrimidine surrogate **5a**, with unsubstituted phenyl at position 4 and H at position 5, was the most potent member studied in this study, with *IC*_50_ of 7.84, 11.51, and 3.84 µM against HepG2, HeLa, and MCF7 cells, respectively. In contrast, compounds with substituted phenyl moiety, such as compound **5c** (4-OCH_3_-Ph), exhibited lower activity against HepG2 and HeLa cells, with *IC*_50_ of 24.39 and 29.76 µM, respectively. However, compound **5c** maintained a strong activity against MCF-7 cells, with an *IC*_50_ of 16.13 µM. The introduction of a trifluoromethyl group at the *para* position of the phenyl moiety, as in compound **5d**, resulted in a strong activity against HepG2 and MCF7 cells, with *IC*_50_ of 11.72 and 8.72, respectively. Nonetheless, a decrease in activity, with an *IC*_50_ of 22.21 µM, was observed against HeLa cells. Replacement of the phenyl moiety with a heterocyclic aromatic ring (pyridinyl and pyrimidinyl), as in compounds **5h**–**j**, led to a diminished activity against all the tested cell lines, except for compound **5h** with a 2-pyridinyl moiety, which displayed a strong activity only against MCF7, with an *IC*_50_ of 14.29 µM. Moreover, the replacement of the H atom at position 5 with CN, as in compounds **5k** and **5l**, resulted in diminished activity against all the tested cell lines. Notably, removal of the sulfonamide moiety from compound **5a** and the insertion of CN in place of H at position 5, as in compound **5m**, resulted in a decrease in activity against all cell lines, except HeLa, which showed very strong growth inhibition, with an *IC*_50_ of 9.83 µM.

To assess the therapeutic safety of the investigated compounds, their cytotoxic activity in the normal fibroblast cell line, WI-38, was evaluated. As illustrated in Table 1, all the tested compounds showed lower cytotoxicity against normal WI-38 cells, as evident from their *IC*_50_ values. Accordingly, the most potent compounds, **3b**, **3f**, **5a**, **5d**, and **5m**, displayed a lower toxic effect on WI-38 cells, with *IC*_50_ values of 93.87, 55.18, 37.90, 51.93, and 73.26, respectively, compared to DOX (*IC*_50_ = 6.72 μM).

### 2.3. CDK7, 8, and 9 Inhibitory Activities

The target compounds **5a**–**m** were assessed for their in vitro CDK7, 8, 9/cyclin T1 enzyme inhibition assay. Atuveciclib (BAY1143572) was selected for comparison as a positive drug control. Dose-response curves were used to calculate *IC*_50_ values (µM), and they are listed in Table 2. As evident from the results of the CDK7 inhibitory assay, compound **5f** is the most potent CDK7 inhibitor with *IC*_50_ of 0.479 µM, compared to other compounds for which the *IC*_50_ ranges from 0.661 to 4.331 µM. Based on the results of the CDK8 inhibitory assay, compound **5d** is the most potent CDK8 inhibitor with an *IC*_50_ of 0.716 µM, compared to other compounds for which the *IC*_50_ ranges from 1.426 to 6.556 µM. Moreover, the results of the CDK9 inhibitory assay revealed that compound **5b** is the most potent CDK9 inhibitor with an *IC*_50_ of 0.059 µM, compared to other compounds for which the *IC*_50_ ranges from 0.073 to 1.957 µM. In addition, compound **5d** exhibited a remarkably higher inhibitory activity against CDK9 with an *IC*_50_ of 0.073 µM than that of atuveciclib (*IC*_50_ = 0.013 µM).

### 2.4. Molecular Docking Studies

As most of the synthesized compounds showed remarkable effects on CDK9, molecular docking studies were performed to obtain insights into the binding of the discovered compounds with CDK9 using the crystal, which was retrieved from the RCSB Protein Data Bank (PDB: 4BCJ) and was co-crystalized with a pyrimidine ligand. Atuveciclib was used as a reference compound to compare the binding pattern. Compounds **5b** and **5d** were selected for molecular docking studies. Both the compounds interact with the ATP active site, suggesting that they act as competitive inhibitors (Figure 3 and Figure 4). It contains an important hinge region and hydrophobic pockets. In the hinge region, interaction with the Cys106 residue is critical for achieving reasonable inhibitory activity against this type of kinase. Asp104 is also an important amino residue located in the hinge region; some CDK9 inhibitors can form dual hydrogen bonds with Cys106 and hydrophobic interactions with Asp104 and other residues such as glycine in the catalytic loop of the ATP active site [40].

For both compounds **5b** and **5d**, the docked poses were aligned to the co-crystallized ligand T9N but showed a slightly different interaction profile, which could explain the similar docking score, as shown in Table 3 and in agreement with the *IC*_50_ determined experimentally. Compound **5b** forms dentate hydrogen bonds with Cys106 and has hydrophobic interaction with the Asp104 residues located in the hinge loop and with Ile25 and Gly26 in the glycine-rich loop, but it did not interact with Asp167 in the catalytic loop; it also interacted with Lys48, a similar interaction that was reported between the phosphate group in ATP and this residue. In the case of **5d**, it exerted the same interactions, but it did not form hydrogen bonds with Asp104 and formed a hydrogen bond with Val33 in the glycine loop. It is worth noting that the interaction profiles of these two compounds were similar to those of the known inhibitor, atuveciclib.

According to the molecular docking studies, it is clear that the 2-anilinopyrimidine moiety is essential for binding. The phenyl group at C4 of the pyrimidine ring, on the other hand, seems to play a role in determining selectivity towards CDK9 as well as potency. In addition, the nature of the substituent on the phenyl ring may have an additional effect on the selectivity. A chlorine atom at the para position of the phenyl ring gave compound **5b**, which was the most potent CDK9 inhibitor. This suggests that the area occupied by such groups may accommodate substituents such as halogens and, to a lesser extent, halogenated methyl groups such as chlorine and trifluoromethyl, respectively. However, a definite structure–activity relationship is not possible until more derivatives are screened.

### 2.5. Molecular Dynamics (MD) Simulation

The previous research suggests that MD simulation, coupled with MM-PBSA, can simulate the inhibitor-binding affinities under physiological settings [41,42,43]. In this study, molecules with lower CDK9 docking scores than that of the (T9N) co-crystal of (4bcj) were confirmed as promising CDK9 inhibitors with superior inhibitory action and greater selectivity. MD simulations may indicate the physical movements of the protein and ligand complex in specific settings to confirm docking results and to compute binding free energies [44]. Because compound **5b** showed remarkable CDK9 activity, it was chosen for MD simulation. This docking complex was subjected to a 50 ns MD simulation run in an explicit hydration environment to assess the binding stability of **5b** and to compare it with the stability of the backbone protein.

#### 2.5.1. Root-Mean-Square Deviation (RMSD)

The RMSD values of the MD simulation trajectories were investigated. As shown in Figure 5, the average RMSD values of **5b** were below 3 Å, indicating that they all remained in good balance throughout the course of the MD simulation. In general, a system is thought to be stable if the RMSD trajectories of a protein–compound complex converge and the average RMSD value remains below 3 Å throughout the MD simulation [45]. On the basis of the preceding results, **5b** exhibited excellent binding stability with CDK9, suggesting that their trajectories could be utilized for further calculations of the binding free energy.

#### 2.5.2. Root-Mean-Square Fluctuations (RMSF)

An analysis of the backbone RMSF values and RMSF charts revealed differences at the atomic level. The RMSF values of the CDK9/cyclinT1 receptor and CDK9/**5b** complex are shown (Figure 6). The collected data from atomic fluctuations were almost exactly the same for both the free receptor (CDK9/cyclinT1) and the complex with low RMSF values. The RMSF plot provided evidence that binding of the compound under consideration to the receptors is stable and does not substantially impact the flexibility of the receptor.

#### 2.5.3. Radius of Gyration (Rg)

The radius of gyration (Rg) data were applied in the simulation to assess if the complex was stable. The Rg value of the protein in complex with **5b** and that of protein alone were, on average, approximately 2.78 and 2.75 nm, respectively, for the course of the dynamics simulation run for 50 ns. This demonstrated that the protein maintained its structure during the MD simulations (Figure 7). According to the Rg plot, the CDK9 receptor and CDK9/**5b** complex exhibited comparable characteristics. Compared to that for the free protein, the Rg values of the protein and protein–compound complexes indicated a high degree of structural similarity and stability.

#### 2.5.4. Hydrogen Bond Analysis

A hydrogen bond analysis is used to understand drug molecular interactions, molecular recognition, and selectivity within the receptors. The study of the hydrogen bonds formed in protein–ligand interactions during MD simulation could be useful for assessing the stability of these interactions. The model for receptor orientation for interacting with the ligand was determined using an MD simulation. For the studied complex, eight hydrogen bonds were formed during MD simulations (Figure 8A). During MD simulations, compound **5b** formed 34 hydrogen bonds with the surrounding residues, such as Asp167, Gly26, Val33, Gln27, Aal153, Phe103, Ala166, Phe103, Lys48, Asp167, Phe105, Asp167, Gly31, Glu66, Asn154, and Glu107 (Figure 8B); however, we picked hydrogen bonds with an occupancy of more than 1%. Therefore, the complex CDK9/**5b** was able to maintain sustained contact with the binding pocket of CDK9 throughout the duration of the simulation. Further investigation showed that CDK9 had strong hydrogen bond interactions with **5b**. The hydrogen bond analysis also explained how **5b** mediated the rotational movement of the hinge region, G-loop, and catalytic loop. As shown in Figure 6B, **5b** exhibits strong proximal-site interactions with surrounding residues, such as Asp104, Asp109, Cys106, Ile25, Thr29, and Phe30, with high occupancy percentages.

#### 2.5.5. Binding Free Energy Calculation

The binding-free energy calculation, based on the chosen MD simulation trajectories, was performed to comprehend the nature of the ligand–protein interaction and to acquire additional information about the contribution of each ligand [46]. In this regard, the MD-directed molecular mechanics Poisson Boltzmann surface area (MM/PBSA) technique was adopted for binding free energy calculation using the ambertool20 mmpbsa tool, wherein a higher negative binding energy explains a greater affinity of a ligand for its target protein. This method can account for a more precise ligand–protein affinity than static or even the most advanced flexible molecular docking methods.

The MM/PBSA technique has an accuracy equivalent to that of the free-energy perturbation techniques, and that too at substantially lower computing costs [47]. Using the SASA-only model of the free-binding energy calculation and the single-trajectory technique, the sample frames were extracted or saved from the final 10 ns of MD simulation trajectories for use in computing each energy term and their average values throughout the three MD simulation runs.

The MM/PBSA method for calculating the binding free energy is highly correlated with experimental activity. In this study, the binding free energies of **5b** were derived from the final 10 ns of the MD trajectories using the MM/PBSA technique. As shown in Table 2, **5b** (−77.56 kcal/mol) exhibited a lower binding free energy. To establish which energy term contributed more to the binding affinity, the four binding free energies (G_vdw_, ΔG_solvation_; Polar, ΔG_solvation_; SASA, and G_ele_) were decomposed into their constituent energy components. As shown in Table 4, van der Waals interactions (G_vdw_) were the most favorable energy providers inside the CDK9–**5b** complex.

### 2.6. Prediction of Physicochemical Properties

#### 2.6.1. Lipinski’s Rule of Five

The Lipinski’s rule of five was used to estimate the drug-likeness properties of compounds **5a**–**m**. Lipinski formulated his rule using a few basic molecular descriptors, such as molecular weight (MW), logP (partition coefficient), and hydrogen bond donor (HBD) and acceptor (HAD) counts, in a molecule to predict the potential oral bioavailability and drug-likeness criteria for the molecules. As illustrated in Table 5, the obtained results indicated that all the investigated compounds, **5a**–**m**, were in accordance with Lipinski’s rule of five with zero violations, indicating that all the compounds had reasonable drug-likeness with acceptable physicochemical properties and could be used as good orally absorbed anticancer agents.

#### 2.6.2. Ligand Efficiency (*LE*) and Liga nd Lipophilic Efficiency (*LLE*)

Ligand efficiency is an important parameter that is widely used in the development of drug measures to quantify the molecular properties, particularly the size and lipophilicity of small molecules, required to gain binding affinity to a drug target [48,49]. The *LE* values of the synthesized compounds were measured using DataWarrior employing the following equation [48]:$$LE=\frac{-RTln{IC}_{50}}{N}$$
where *R* is the universal gas constant, *T* is the absolute temperature in Kelvin, and *N* represents the number of heavy atoms, that is, non-hydrogen atoms in the drug.

*LE* is widely used for the selection and optimization of fragments, hits, and leads. In particular, the optimization of lipophilic ligand efficiency demonstrates that it is possible to increase the affinity and decrease the lipophilicity simultaneously, even with challenging “lipophile-preferring” targets. Compounds with *LE* values higher than 0.3 are promising lead compounds. The *LE* values for the target compounds in this study are presented in Table 6, which shows that the *LE* values of the synthesized compounds are between 0.3 and 0.4.

##### Ligand Lipophilic Efficiency (*LLE*)

*LLE* is a widely used parameter in drug design and discovery to estimate the quality of target compounds, linking potency, and lipophilicity to evaluate drug-likeness [50].

In drug discovery and development, it is challenging to optimize the compound activity while maintaining a constant lipophilicity. As a result, *LLE* is an effective approach to control molecule lipophilicity to evade any “molecular obesity” through lead optimization. The *LLE* values of these compounds are shown in Table 6. These were obtained via DataWarrior, according to the following equation [50]:$$LLE=p{IC}_{50}-clogP$$

An acceptable lead compound should have an *LLE* value ≥5. As depicted in Table 6, **5j** has acceptable *LLE* values. However, other compounds have values below the recommended limit. In conclusion, **5j** is a good candidate for lead optimization because it has a non-hydrogen atom (*N*) content of 23, an acceptable *LE* value of 0.38673, and an acceptable *LLE* value of 5.5526.

#### 2.6.3. ADMET Prediction

In drug development, plasma protein binding (PPB), which shows the extent to which a drug is bound to the plasma, is a trait that needs to be given special attention. As only a free drug can elicit a pharmacological reaction [51], the amount of a drug bound to proteins may affect its efficacy and toxicity. Human intestinal absorption (HIA) refers to the process by which oral medications are absorbed into the bloodstream from the digestive tract [52]. All the aforementioned characteristics are crucial indicators of the potential safety and pharmacological activity of the selected compounds, particularly those used to treat cancer [52].

Considering the lack of success with the majority of the previously reported CDK9 inhibitors in clinical trials owing to several substantial adverse effects, it is vital to assess the absorption, distribution, metabolism, and excretion (ADMET) characteristics during the early stages of drug development to avoid failure. In this study, the ADMET properties of 15 studied compounds, including atuveciclib as a reference, were predicted using the admetSAR server (Table 7).

With regard to absorption, the results of HIA and human oral bioavailability (HOB) indicated that all the hits, including atuveciclib, can potentially be used as oral medications. For distribution, the 15 studied compounds showed considerable plasma protein-binding properties, indicating that they are difficult to be displaced by other medicines from plasma proteins into the blood.

Each of these compounds can cross the blood–brain barrier (BBB), which is essential for the treatment of cancer that has spread to the brain. The effectiveness and quantity of CYP2D6, a drug-metabolizing enzyme, produced by individuals varies considerably [53]. As shown in Table 7, none of the investigated compounds were affected by CYP2D6, indicating that they may have consistent metabolic rates within the human body. All the compounds were non-carcinogenic and exhibited mild hepatotoxicity, indicating their safety for human consumption. In general, all the compounds exhibited favorable ADMET characteristics, indicating their excellent potential as anticancer therapeutic candidates.

## 3. Experimental

### 3.1. Chemistry

All chemical reagents and solvents used in the synthesis of all target compounds were obtained from commercial suppliers, Apollo Scientific (Manchester, UK), Ark Pharm (Arlington Heights, IL, USA), and Cambridge Isotope Laboratories (Tewksbury, MA, USA), and were used directly without further purification. All reactions were monitored using thin-layer chromatography (TLC) on glass sheets (silica gel F254) and visualized under UV light. An Agilent 6320 Ion Trap mass spectrometer was used to generate the mass spectra (MS). The Melting Point Apparatus Barnstead 9100 Electrothermal was used to record the melting points of the final compounds. Infrared (IR) spectra were obtained using an FT-IR spectrometer (Perkin-Elmer, Waltham, MA, USA). A Bruker 700 Ultrashield NMR spectrometer was run at 700 and 175 MHz to generate ^1^H and ^13^C spectra, respectively.

#### 3.1.1. General Procedure for the Preparation of Enaminone Derivatives (**2a**–**m**)

The synthesis started with the condensation of acetopheneone derivatives (1 eq, 6.88 mmol) (**1a**–**m**) and *N, N*-dimethylformamide dimethyl acetal (DMF-DMA) (2 eq, 13.76 mmole) to generate the corresponding enaminones (**2a**–**m**). The crude product, thus, obtained was used in the next step without further purification [18,19].

#### 3.1.2. General Procedure for the Preparation of 2-Aminopyrimidine Derivatives (**3a**–**h**)

In the solution of ((*E*)-3-(dimethylamino)-1-phenylprop-2-en-1-on (0.1 g, 0.57 mmol), 3-guanidinonitrate (2 eq), the K_2_CO_3_ (2 eq) reaction mixture was added in 5 mL of *n*-butanol. The reaction mixture was then stirred and refluxed overnight. After consumption, the mixture was cooled to room temperature and poured into water to obtain a white precipitate.

*4-phenylpyrimidin-2-amine* (**3a**), White solid (0.085 g, 87%). M.p. 160–161 °C. IR (*ν*max/cm^−1^): 3302 (NH), 3146 (CH), 1651, 1551 (C=N and C=C), 1211 (C–N). ^1^H NMR (DMSO-*d*_6_) δ 8.31 (d, *J* = 5.1 Hz, 1H), 8.10–8.04 (m, 2H), 7.50–7.49 (m, 3H), 7.12 (dd, *J* = 5.2, 1.2 Hz, 1H), 6.66 (s, 2H, NH_2_). ^13^C NMR (DMSO-*d*_6_) δ 164.8, 164.7, 159.7, 137.5, 130.9, 129.2, 127.2, 106.3. MS, *m*/*z* (%): 171.93 [M + 1].*4-(4-chlorophenyl)pyrimidin-2-amine* (**3b**), White solid (0.095 g, 81%). M.p.127-129 °C. (lit. 128.46 °C [53]) IR (*ν*max/cm^−1^): 3459 (NH), 3053 (CHs), 1620, 1542 (C=N and C=C), 1122 (C–N), 796 (C–Cl). ^1^H NMR (DMSO-*d*_6_) δ 8.32 (d, *J* = 4.6 Hz, 1H), 8.11–8.06 (m, 2H), 7.56 (dd, *J* = 8.4, 2.8 Hz, 2H), 7.15–7.12 (m, 1H), 6.71 (s, 2H, NH_2_). ^13^C NMR (DMSO-*d*_6_) δ 164.3, 162.8, 159.8, 136.3, 135.7, 129.2, 128.9, 106.1. MS, *m*/*z* (%): 206.03 [M + 1].*4-(4-methoxyphenyl)pyrimidin-2-amine* (**3c**), White solid (0.098 g, 85%). M.p. 189-191 °C. (lit 191 °C) [54] IR (*ν*max/cm^−1^): 3456 (NH), 3130, 2989 (CHs), 1614, 1548 (C=N and C=C), 1170 (C-O). ^1^H NMR (DMSO-*d*_6_) δ 8.24 (d, *J* = 5.2 Hz, 1H), 8.04 (d, *J* = 8.8 Hz, 2H), 7.06 (d, *J* = 5.2 Hz, 1H), 7.04 (d, *J* = 8.8 Hz, 2H), 6.57 (s, 2H, NH_2_), 3.82 (s, 3H, OCH_3_). ^13^C NMR (DMSO-*d*_6_) δ 164.2, 163.6, 161.7, 159.2, 129.8, 128.7, 114.5, 105.5, 55.8. MS, *m*/*z* (%): 201.96 [M + 1].*4-(4-(trifluoromethyl)phenyl)pyrimidin-2-amine* (**3d**), White solid (0.118 g, 86%). M.p. 180–182 °C. IR (*ν*max/cm^−1^): 3484 (NH), 3156 (CH), 1627, 1553 (C=N and C=C), 1311 (CF_3_), 1120 (C–N). ^1^H NMR (DMSO-*d*_6_) δ 8.38 (d, *J* = 5.1 Hz, 1H), 8.29–8.25 (m, 2H), 7.87 (d, *J* = 8.2 Hz, 2H), 7.21 (d, *J* = 5.1 Hz, 1H), 6.80 (s, 2H, NH_2_). ^13^C NMR (DMSO-*d*_6_) δ 164.33, 162.49, 160.07, 141.42, 130.35 (q,^2^*J*_F-C_ = 33.25 Hz), 127.95, 126.10 (q,^3^*J*_F-C_ = 3.50 Hz), 124.61 (q,^1^*J*_F-C_ = 271.25 Hz), 106. 73.MS, *m*/*z* (%): 240.01 [M + 1].*4-(4-(trifluoromethoxy)phenyl)pyrimidin-2-amine* (**3e**), White solid (0.122 g, 85%). M.p. 188–190 °C. IR (*ν*max/cm^−1^): 3280 (NH), 3136, 2931 (CHs), 1625, 1560 (C=N and C=C), 1278 (CF_3_), 1150 (C–O). ^1^H NMR (DMSO-*d*_6_) δ 8.34 (d, *J* = 5.1 Hz, 1H), 8.19 (d, *J* = 8.8 Hz, 2H), 7.52–7.43 (m, 2H), 7.15 (d, *J* = 5.2 Hz, 1H), 6.73 (s, 2H, NH_2_). ^13^C NMR (DMSO-*d*_6_) δ 163.80, 162.14, 159.35, 149.84, 136.19, 128.79, 121.03 (q,^1^*J*_F-C_ = 255.50 Hz), 120.04, 105.85. MS, *m*/*z* (%): 256.02 [M + 1].*2-amino-4-phenylpyrimidine-5-carbonitrile* (**3f**), White solid (0.098 g, 88%). M.p. 149-151 °C. (lit. 150.85 °C [55]), IR (*ν*max/cm^−1^): 3290 (NH), 3127 (CHs), 2210 (CN), 1656, 1572 (C=N and C=C), 1214 (C–N). ^1^H NMR (DMSO-*d*_6_) δ 8.73 (s, 1H), 7.88–7.82 (m, 4H), 7.61–7.54 (m, 4H).^1 13^C NMR (DMSO-*d*_6_) δ 168.5, 164.2, 163.8, 136.4, 131.6, 129.0, 128.8, 118.6, 93.1. MS, *m*/*z* (%): 195.04 [M + 1].*2-amino-4-(4-chlorophenyl)pyrimidine-5-carbonitrile* (**3g**), White solid; (0.1 g, 76%). M.p. 166-168 °C. (lit. 167.60 °C [56]), IR (*ν*max/cm^−1^): 3297 (NH), 3107 (CH), 2205 (CN), 1647, 1578 (C=N and C=C), 1082 (C–N), 786 (C−Cl). ^1^H NMR (DMSO-*d*_6_) δ 8.74 (s, 1H), 7.93–7.85 (m, 4H), 7.65 (d, *J* = 8.5 Hz, 2H). ^13^C NMR (DMSO-*d*_6_) δ 167.3, 164.3, 163.8, 136.5, 135.1, 130.6, 129.2, 118.4, 93.0. MS, *m*/*z* (%): 231.65 [M − 1].*2-amino-4-(4-fluorophenyl)pyrimidine-5-carbonitrile* (**3h**), White solid (0.107 g, 88%). M.p. 214–216 °C. IR (*ν*max/cm^−1^): 3290 (NH), 3106 (CHs), 2216 (CN), 1653, 1571 (C=N and C=C), 1228 (C−F), 1157 (C−N). ^1^H NMR (DMSO-*d*_6_) δ 8.73 (s, 1H), 7.94 (dd, *J* = 8.7, 5.5 Hz, 2H), 7.86 (d, *J* = 14.0 Hz, 2H), 7.44–7.37 (m, 3H). ^13^C NMR (DMSO-*d*_6_) δ 167.3, 164.2 (d,^1^*J*_F-C_ = 248.5 Hz), 164.24, 163.74, 132.82 (d,^4^*J*_F-C_ = 3.5 Hz), 131.36 (d,^3^*J*_F-C_ = 8.75 Hz), 118.5, 116.14 (d,^2^*J*_F-C_ = 21 Hz), 92.93. MS, *m*/*z* (%): 212.89 [M − 2].

#### 3.1.3. General Procedure for the Preparation of Guanidines (**4a**,**b**)

The synthesis started from the commercially available 3-aminobenzene or 3-amino-benzenesulfonamide (0.2 gm, 1.1 mmol), which was weighed and placed in a 50 mL round-bottom flask equipped with a magnetic stir bar; cyanamide (4 eq) and trimethylsilyl (2.2 eq) were added in 10 mL of acetonitrile. The flask was fitted with a cold-water condenser and heated at reflux with constant stirring and heated to 75 °C for 24 h, after which it was allowed to cool to room temperature and showed the onset of precipitation. The beaker was cooled in an ice bath to complete the precipitation of the product. If required, the bottom of the beaker was gently scratched to induce crystallization. The product was collected via vacuum filtration, washed with a small amount of ice-cold water, and air dried. A small sample of the crude product was saved, and the remainder was purified through recrystallization by dissolving in a minimum amount of boiling hot water, allowing the solution to slowly cool to room temperature, and then further cooling in an ice bath to crystallize as much of the product from the solution as possible. The recrystallized product was collected via vacuum filtration to obtain a white product. The product guanidine (**4a**,**b**) was weighed, and the percentage yield was calculated. The calculated melting point was similar to that reported for guanidines [57].

*1-Phenylguanidine* (**4a**), ^1^H NMR (DMSO-*d*_6_) δ 7.56 (d, *J* = 8.7 Hz, 2H), 6.79 (dd, *J* = 8.7, 2.3 Hz, 2H), 6.69 (d, *J* = 2.3 Hz, 2H), 6.10 (d, *J* = 1.5 Hz, 2H).*3-Guanidinobenzenesulfonamide* (**4b**), Tan white solid (0.2 g, 85%).M.p 164–166 (lit. 165.13 °C [58]). ^1^H NMR (DMSO-*d*_6_) δ 10.26 (s, 1H), 7.72–7.69 (m, 4H), 7.67–7.61 (m, 2H), 7.47 (d, J = 6.9 Hz, 3H). ^13^C NMR (DMSO-*d*_6_) δ 155.95, 145.30, 136.13, 130.48, 127.21, 123.11, 120.95

#### 3.1.4. General Procedure for the Preparation of 2-Anilinopyrimidine Derivatives (**5a**–**m**)

In the solution of ((*E*)-3-(dimethylamino)-1-phenylprop-2-en-1-on (0.1 gm, 0.57 mmol), 3-guanidinoderivative (0.24 gm, 1.14 mmol), the K_2_CO_3_ (0.15 gm, 1.14 mmol) reaction mixture was added in 5 mL of *n*-butanol. The reaction mixture was then stirred and refluxed overnight. After consumption, the mixture was cooled to room temperature and poured into water to obtain a white precipitate.

*3-((4-Phenylpyrimidin-2-yl)amino)benzenesulfonamide* (**5a**), White solid (0.159 g, 85%). M.p. >300 °C. IR (*ν*max/cm^−1^): 3271 (NH), 2931 (CHs), 1634, 1556 (C=N and C=C), 1410 (S=O), 1143 (C−N). ^1^H NMR (DMSO-*d*_6_) δ 9.90 (s, 1H), 8.60–8.53 (m, 1H), 8.51 (s, 1H), 8.27–8.22 (m, 3H), 7.79 (s, 1H), 7.58–7.51 (m, 5H), 7.48–7.37 (m, 2H). ^13^C NMR (DMSO-*d*_6_) δ 163.57, 159.97, 159.22, 136.36, 131.03, 128.39, 127.10, 118.34, 115.67. MS, *m*/*z* (%): 325.11 [M − 1]*3-((4-(4-Chlorophenyl)pyrimidin-2-yl)amino)benzenesulfonamide* (**5b**), White solid (0.180 g, 87%). M.p. 245–247 °C. White solid (0.170 g, 83%). M.p. 263–265 °C. IR (*ν*max/cm^−1^): 3263 (NH), 3070 (CH), 1567, 1543 (C=N and C=C), 1425 (S=O), 1145 (C-N), 773 (C−Cl). ^1^H NMR (DMSO-*d*_6_) δ 10.08 (s, 1H, NH), 8.67–8.66 (m, 1H), 8.63 (d, *J* = 5.2 Hz, 1H), 8.28 (d, *J* = 8.7 Hz, 2H), 7.84-7.86 (m, 1H), 7.61 (d, *J* = 8.6 Hz, 2H), 7.54–7.49 (m, 2H), 7.45 (dt, *J* = 7.8, 1.3 Hz, 1H), 7.33 (s, 2H, NH_2_). ^13^C NMR (DMSO-*d*_6_) δ 162.81, 160.36, 160.00, 144.94, 141.36, 136.31, 135.65, 129.64, 129.45, 129.37, 122.15, 118.91, 116.17, 108.90^.^MS, *m*/*z* (%): 358.91[M − 2].*3-((4-(4-Methoxyphenyl)pyrimidin-2-yl)amino)benzenesulfonamide* (**5c**), A white solid (0.117 g, 87%) was obtained. M.p. 242–244 °C. IR (νmax/cm^−1^): 3270 (NH), 2930 (CHs), 1652, 1590 (C=N and C=C), 1422 (S=O), 1253 (C−O), 1168 (C−N). ^1^H NMR (DMSO-d_6_) δ 9.96 (s, 1H), 8.66 (d, *J* = 2.3 Hz, 1H), 8.53 (d, *J* = 5.2 Hz, 1H), 8.22 (d, *J* = 8.8 Hz, 2H), 7.87–7.82 (m, 1H), 7.49 (s, 1H), 7.43 (d, *J* = 5.4 Hz, 2H), 7.32 (s, 2H), 7.08 (d, *J* = 8.8 Hz, 2H), 3.85 (s, 3H, OCH_3_). ^13^C NMR (DMSO-d_6_) δ 163.7, 162.2, 160.3, 159.3, 145.0, 141.6, 129.3, 129.2, 129.1, 122.0, 118.7, 116.0, 114.7, 108.2, 55.9. MS, *m*/*z* (%): 355.18 [M − 1].*3-((4-(4-(Trifluoromethyl)phenyl)pyrimidin-2-yl)amino)benzenesulfonamide* (**5d**), White solid (0.195 g, 87%). M.p. 150–152 °C. IR (*ν*max/cm^−1^): 3297, 3207 (NHs), 3070 (CH), 1545 (C=C), 1420 (S=O), 1319 (C-F), 1111 (C−N). ^1^H NMR (DMSO-*d*_6_) δ 10.15 (s, 1H, NH), 8.72–8.63 (m, 2H), 8.45 (d, *J* = 7.9 Hz, 2H), 7.90 (d, *J* = 8.0 Hz, 2H), 7.87–7.82 (m, 1H), 7.60 (d, *J* = 5.2 Hz, 1H), 7.50 (d, *J* = 7.9 Hz, 1H), 7.44 (d, *J* = 7.7 Hz, 1H), 7.34 (s, 2H, NH_2_). ^13^C NMR (DMSO-*d*_6_) δ 162.4, 160.4, 160.3, 145.0, 141.3, 140.8, 132.3 (q,^2^*J*_F-C_ = 29.75 Hz), 129.7, 128.4, 126.26 (q,^3^*J*_F-C_ = 3.5 Hz), 124.05 (q,^1^*J*_F-C_ = 269.5 Hz), 122.2, 119.0, 116.3, 109.5. MS, *m*/*z* (%): 393.02 [M − 1].*3-((4-(4-(Trifluoromethoxy)phenyl)pyrimidin-2-yl)amino)benzenesulfonamide* (**5e**), White solid; (0.2 g, 86%). M.p. 252–254 °C. IR (*ν*max/cm^−1^): 3349, 3288 (NHs), 3072 (CH), 1549 (C=C), 1424 (S=O), 1277 (C–F), 1153 (CFO). ^1^H NMR (DMSO-*d*_6_) δ 10.10 (s, 1H, NH), 8.68–8.61 (m, 2H), 8.41–8.33 (m, 2H), 7.85 (dd, *J* = 8.3, 2.2 Hz, 1H), 7.56–7.49 (m, 4H), 7.47–7.41 (m, 1H), 7.34 (s, 2H, NH_2_). ^13^C NMR (DMSO-*d*_6_) δ 162.6, 160.4, 160.1, 150.7, 145.0, 141.4, 136.0, 129.7, 129.6, 122.2, 121.6, 121.2, 120.04 (q,^1^*J*_F-C_ = 257.14 Hz), 118.9, 116.2, 109.1. MS, *m*/*z* (%): 411.07 [M − 1].*3-((4-(4-Fluorophenyl)pyrimidin-2-yl)amino)benzenesulfonamide* (**5f**), White solid (0.167 g, 86%). M.p. 285–287 °C. IR (*ν*max/cm^−1^): 3296 (NH), 3109 (CH), 1590, 1562 (C=N and C=C), 1418 (S=O), 1231 (C–F), 1147 (C–N).^1^H NMR (DMSO-*d*_6_) δ 10.05 (s, 1H, NH), 8.66 –8.65(m, 1H), 8.61 (d, *J* = 5.2 Hz, 1H), 8.34–8.30 (m, 2H), 7.86 (m, 1H), 7.53–7.48 (m, 2H), 7.46–7.42 (m, 1H), 7.40–7.31 (m, 4H). ^13^C NMR (DMSO-*d*_6_) δ 164.42 (d,^1^*J*_F-C_ = 246.75 Hz), 162.97, 160.35, 159.81, 144.98, 141.42, 133.32 (d,^4^*J*_F-C_ = 3.5 Hz), 130.03 (d,^3^*J*_F-C_ = 8.75 Hz), 129.61, 122.10, 118.85, 116.33 (d,^2^*J*_F-C_ = 22.75 Hz), 116.16, 108.77.MS, *m*/*z* (%): 341.08 [M − 2].*3-((4-(4-Nitrophenyl)pyrimidin-2-yl)amino)benzenesulfonamide* (**5g**), White solid (0.180 g, 85%). M.p. 265–267 °C. IR (*ν*max/cm^−1^): 3465, 3391 (NHs), 3015, 2916 (CHs), 1590, 1515 (C=N and C=C), 1550, 1331 (NO_2_), 1414 (S=O), 1151 (C–N). ^1^H NMR (DMSO-*d*_6_) δ 10.20 (s, 1H, NH), 8.72 (d, *J* = 5.1 Hz, 1H), 8.68 (s, 1H), 8.52–8.49 (m, 2H), 8.37 (d, *J* = 8.8 Hz, 2H), 7.84–7.81 (m, 1H), 7.64 (d, *J* = 5.1 Hz, 1H), 7.51 (m, 1H), 7.47–7.44 (m, 1H), 7.36 (s, 2H, NH_2_). ^13^C NMR (DMSO-*d*_6_) δ 161.7, 160.6, 160.4, 149.3, 145.0, 142.8, 141.2, 129.7, 128.9, 124.5, 122.3, 119.1, 116.3, 109.9. MS, *m*/*z* (%): 370.04 [M − 1].*3-((4-(Pyridin-2-yl)pyrimidin-2-yl)amino)benzenesulfonamide* (**5h**), White solid (0.160 g, 86%). M.p. 247–249 °C. IR (*ν*max/cm^−1^): 3266 (NH), 3000 (CH), 1590, 1537 (C=N and C=C), 1418 (S=O), 1151 (C–N). ^1^H NMR (DMSO-*d*_6_) δ 10.13 (s, 1H, NH), 8.75 (m, 1H), 8.71 (s, 1H), 8.69 (d, *J* = 5.0 Hz, 1H), 8.53 (d, *J* = 7.9 Hz, 1H), 8.01–7.99 (m, 1H), 7.85 (dd, *J* = 8.1, 2.2 Hz, 1H), 7.80 (d, *J* = 5.0 Hz, 1H), 7.58 (m,1H), 7.53–7.51 (m, 1H), 7.46–7.43 (m, 1H), 7.34 (s, 2H, NH_2_). ^13^C NMR (DMSO-*d*_6_) δ 163.3, 160.3, 160.3, 153.9, 150.1, 145.0, 141.4, 138.1, 129.7, 126.4, 122.1, 121.9, 118.9, 116.2, 109.0. MS, *m*/*z* (%): 325.97 [M − 2].*3-((4-(Pyridin-4-yl)pyrimidin-2-yl)amino)benzenesulfonamide* (**5i**), White solid (0.158 g, 85%). M.p. 289–291 °C. IR (*ν*max/cm^−1^): 3266 (NH), 3023 (CH), 1590, 1537 (C=N and C=C), 1420 (S=O), 1171 (C–N). ^1^H NMR (DMSO-*d*_6_) δ 10.18 (s, 1H, NH), 8.80–8.75 (m, 2H), 8.72 (d, *J* = 5.1 Hz, 1H), 8.67 (s, 1H), 8.19–8.11 (m, 2H), 7.89–7.83 (m, 1H), 7.63 (d, *J* = 5.1 Hz, 1H), 7.53 –7.50 (m, 1H), 7.47–7.43 (m, 1H), 7.34 (s, 2H, NH_2_). ^13^C NMR (DMSO-*d*_6_) δ 163.3, 160.6, 160.5, 151.1, 145.0, 144.0, 141.2, 129.7, 122.2, 121.5, 119.1, 116.3, 109.6. MS, *m*/*z* (%): 325.97 [M − 2].*3-((4-(Pyrazin-2-yl)pyrimidin-2-yl)amino)benzenesulfonamide* (**5j**), White solid (0.152 g, 81%). M.p. 288–290 °C.IR (*ν*max/cm^−1^): 3282 (NH), 3071 (CHs), 1594, 1567 (C=N and C=C), 1426 (S=O), 1148 (C–N). ^1^H NMR (DMSO-*d*_6_) δ 10.22 (s, 1H, NH), 9.62 (d, *J* = 1.5 Hz, 1H), 8.87–8.79 (m, 2H), 8.75 (d, *J* = 4.9 Hz, 1H), 8.63 (d, *J* = 2.0 Hz, 1H), 7.88 (dd, *J* = 8.2, 2.2 Hz, 1H), 7.76 (d, *J* = 4.9 Hz, 1H), 7.55–7.52 (m, 1H), 7.46 (dt, *J* = 7.9, 1.3 Hz, 1H), 7.38–7.30 (m, 2H). ^13^C NMR (DMSO-*d*_6_) δ 161.7, 160.7, 160.3, 148.9, 147.2, 145.1, 144.9, 143.3, 141.2, 129.7, 122.3, 119.1, 116.2, 109.4. MS, *m*/*z* (%): 324.05 [M − 1].*3-((4-(4-Chlorophenyl)-5-cyanopyrimidin-2-yl)amino)benzenesulfonamide* (**5k**), White solid (0.198 g, 90%). M.p. >300 °C. IR (*ν*max/cm^−1^): 3311, 3243 (NHs), 3104 (CH), 2221 (CN), 1584, 1554 (C=N and C=C), 1431 (S=O), 1157 (C–N), 787 (C–Cl). ^1^H NMR (DMSO-*d*_6_) δ 9.01 (s, 1H), 8.55–8.31 (m, 1H), 8.05 (s, 2H), 7.86 (d, *J* = 6.8 Hz, 1H), 7.68 (d, *J* = 8.2 Hz, 3H), 7.54 (d, *J* = 6.5 Hz, 3H). ^13^C NMR (DMSO-*d*_6_) δ 167.7, 164.3, 159.9, 145.1, 139.8, 136.7, 134.7, 131.0, 129.9, 129.4, 123.7, 120.8, 117.8, 117.7, 95.6. MS, *m*/*z* (%): 384.06 [M − 1].*3-((5-Cyano-4-(4-fluorophenyl)pyrimidin-2-yl)amino)benzenesulfonamide* (**5l**), White solid (0.173 g, 82%). M.p. >300 °C. IR (*ν*max/cm^−1^): 3307, 3246 (NHs), 2218 (CN), 1562 (C=C), 1430 (S=O), 1233 (C–F), 1156 (C–N). ^1^H NMR (DMSO-*d*_6_) δ 10.82 (s, 1H, NH), 9.01 (s, 1H), 8.44 (s, 1H), 8.11 (s, 2H), 7.88 (d, *J* = 7.4 Hz, 1H), 7.59–7.48 (m, 2H), 7.46–7.44 (m, 2H), 7.38 (s, 2H, NH_2_). ^13^C NMR (DMSO-*d*_6_) δ 164.02 (d,^1^*J*_F-C_ = 248.5 Hz), 159.9, 145.1, 139.7, 132.34 (q,^4^*J*_F-C_ = 2.7 Hz), 131.82 (d,^3^*J*_F-C_ = 8.75 Hz), 129.9, 123.7, 120.8, 117.8, 117.7, 116.38 (d,^2^*J*_F-C_ = 21 Hz). MS, *m*/*z* (%): 368.09 [M − 1].*4-Phenyl-2-(phenylamino)pyrimidine-5-carbonitrile* (**5m**), White solid (0.111 g, 71%). M.p. 210–212 °C. IR (*ν*max/cm^−1^): 3276 (NH), 3102, 3043 (CHs), 2213 (CN), 1552 (C=C), 1218 (C–N). ^1^H NMR (DMSO-*d*_6_) δ 10.53 (s, 1H, NH), 8.97 (s, 1H), 7.98–7.94 (m, 2H), 7.77 (d, *J* = 7.9 Hz, 2H), 7.65–7.60 (m, 3H), 7.36 (dd, *J* = 8.6, 7.3 Hz, 2H), 7.08 (t, *J* = 7.3 Hz, 1H). ^13^C NMR (DMSO-*d*_6_) δ 167.8, 163.4, 159.7, 138.8, 135.7, 131.4, 128.9, 128.7, 128.6, 128.5, 123.4, 120.4, 117.6. MS, *m*/*z* (%): 271.06 [M − 1].

### 3.2. Antitumor Screening

The MTT assay was performed to evaluate the in vitro antitumor activity of the synthesized 4-anilinoquinazolines according to a previously reported method [59,60,61,62,63].

### 3.3. In Vitro CDK Inhibition Assay

The CDK enzyme assay was performed as previously described [63].

### 3.4. Molecular Docking

A molecular docking simulation was performed using the Molecular Operating Environment (MOE) software. First, the 3D crystal structure of CDK9 was retrieved from the RCSB Protein Data Bank (PDB: 4BCJ) [63], and then, it was subjected to the protein preparation protocol in MOE where missing atoms, loops, steric clashes were minimized and the selection of alternative conformations was performed and, finally, 3D protonated. For compound preparation, the MMFF94 force field was used for energy minimization, hydrogen atoms were added, partial charges were estimated, and compounds were saved as mol2 files. Molecular docking into the ATP binding site and calculations were performed using the triangle matcher algorithm, and the docking score between CDK9 and each ligand was computed using the London-dG docking scoring function and rescored with the GBVI/WSA-dG function in MOE [26,44,53,64].

### 3.5. Molecular Dynamics (MD) Simulation

The molecular dynamics simulation was performed using NAMD Git-2021-09-06 Linux-x86 64 multicore [61] and the AMBER force field, and various 50 ns MD simulations were performed to assess the binding stabilities of compound **5b** The CHARMM-GUI website (http://www.charmm-gui.org/ (accessed on 20 February 2023)) was used to produce the MD simulation configuration files. As a physiological condition, the poses that showed the best binding affinity in the molecular docking study were solvated in a 10.0 periodic dodecahedron water box with 0.15 M salt content. The ligand was parameterized using the CHARMM General Force Field (CGenFF) web-based tool (https://cgenff.umaryland.edu/ (accessed on 20 February 2023)) [51,65,66,67,68], and the ligand was parameterized. All the systems were minimized and equilibrated for 200 ps in the NPT and NVT steps.

Each MD simulation run was performed at a time step of 2 fs. The stability of each system was assessed by computing the RMSD over the entire simulation period. Only compounds showing stable binding during the entire MD simulation run, with convergent RMSD values and no major fluctuations [44], were further subjected to the subsequent binding free energy calculations [37].

### 3.6. Binding Free Energy Calculations

The binding free energy (ΔG_bind_) was calculated using the MM-PBSA approach [40]. All energy components, including van der Waals, electrostatic, polar solvation, and nonpolar solvation contributions, were computed using the 1000 snapshots collected from the last 10 ns of the MD trajectories.

### 3.7. ADMET Prediction

The pharmacokinetic and cytotoxic properties of the synthesized compounds were predicted using the admetSAR server (http://lmmd.ecust.edu.cn/admetsar1 (accessed on 20 February 2023)), which assessed their carcinogenicity, ability to interact with CYP2D6, BBB permeability, PPB, and HIA. As a component of the cytochrome P450 mixed-function oxidase system, CYP2D6 is responsible for the metabolism and elimination of drugs [64]. The BBB is a semipermeable border of endothelial cells that prevents drugs in circulation from entering the human central nervous system. This trait has emerged as an essential requirement in the development of new drugs [26].

## 4. Conclusions

A novel series of molecules structurally related to 2-anilinopyrimidine were designed, synthesized, and structurally evaluated for their physicochemical and biological properties, including CDK9 inhibitory, cytotoxic, and antiproliferative activities against human lung fibroblasts (WI-38), hepatocellular carcinoma (HepG2), colorectal carcinoma (HCT116), mammary gland breast cancer (MCF7), and epithelioid carcinoma (HeLa). The results of the CDK inhibitory assay revealed that compound **5b** exhibited a promising effect against CDK9 and was the most potent CDK9 inhibitor (*IC*_50_: 0.059 µM), **5f** was the most potent CDK7 inhibitor (*IC*_50_: 0.479 µM), and **5d** was the most potent CDK8 inhibitor (*IC*_50_: 0.716 µM). Some anilinopyrimidine derivatives, particularly **5a**, **5d**, and **5h**, exhibited very strong potent antiproliferative activity with *IC*_50_ of 1–10 μM in cancer cells. In addition, compound **5b** showed the highest activity against CDK9, which was further assessed using computer modeling platforms.

## Data Availability

Not applicable.

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
