# Peer review of "Targeting Transcriptional CDKs 7, 8, and 9 with Anilinopyrimidine Derivatives as Anticancer Agents: Design, Synthesis, Biological Evaluation and In Silico Studies"

_molecules, 2023, doi:10.3390/molecules28114271_

Round 1

Reviewer 1 Report

The manuscript described the synthesis, biological activities and in silico studies of a series of anilinopyrimidine derivatives. 

There are some questions about the manuscript as follows:

1.  The key word "biological activities" or similar phrase should be emerged in the title, and the phrase may be added between "synthesis" and "and" in the title. 

2.  Authors should describe the adequate illustrations for the choices of the substituents including pyridinyl and pyrimidinyl. 

3.  Authors should describe clearly the basis and rationale of the choices of the five cancer cell lines (HCT116, HepG2, HeLa, MCF7 and WI-38) for biological study of the target compounds, and the relationships between the five cancer cell lines and cyclin-dependent kinases. 

4.   Authors should describe clearly the basis and rationale of the choice of the crystal structure (4BCJ) of protein-ligands for molecular docking of the target compounds. 

5.  As to the multiplet splitting peaks of proton NMR spectra, the value of chemical shift should be a numerical range. For example,  "7.05 (m, 1H)" in compound 5o. 

6.  As known, the measuring solvents in NMR spectra are deuterated DMSO (CD3SOCD3), deuterated acetone (CD3COCD3), etc., not non-deuterated.

7.  There are some errors in attribution for the splitting types and and coupling constant of the proton NMR spectra in aromatic region of the target compounds.  For example, triple peaks of  8.67 (t, J = 2.0 Hz, 1H) in compound 5b, 8.66 (t, J = 2.0 Hz, 1H) in compound 5f, 7.51 (t, J = 7.9 Hz, 1H) in compound 5g, 8.01 (td, J = 7.7, 1.8 Hz, 1H), 7.52 (t, J = 7.9 Hz, 1H) in compound 5h,  7.52 (t, J = 7.9 Hz, 1H) in compound 5k, 7.53 (t, J = 7.9 Hz, 1H) in compound 5l are simply wrong, these triple peaks are impossible. These hydrogen protons of these compounds have appeared as pseudo triplet splittings. Authors should carefully check the proton NMR data of all target compounds.

8.  As known, Ph is the specific symbol for phenyl group, not used in the substituted phenyl group, therefore the descriptions for the substituents in Table 1 are wrong. For example,4-Cl-Ph should be revised to "4-ClC6H4" or should be expressed with the graphic structure.

9.  The codes of the target compounds are poor to be confused, authors should add the numbers before the code letters of the target compounds. For example, "3a‒h" should be changed to "3a‒8h".

10.  There are some errors in use of superscript and subscript. For example, " IC50".

11.  Authors should provide the experimental procedure for synthesis of the intermediates 4a and 4b. 

12.  There are some errors in typos, spelling, syntax, punctuation, usage of capital letters, consistency in language style in the manuscript.

Based on the above, I think this manuscript should be accepted with positive revision for publication in MOLECULES. 

Author Response

Thank you very much for the helpful comments and suggestions.

We have modified the manuscript accordingly and detailed corrections are listed below point by point.

Please, find enclosed my answers to reviewers with the list of changes that have been highlighted in the manuscript in yellow color.

Reviewer 1

  • The title was changed accordingly.
  • The choice of the cell lines was mentioned in section 2.2.
  • The choice of the crystal structure was mentioned in section 2.4.
  • - All of the suggested modifications in comments 4-12 were made accordingly

Reviewer 2

  • The docking details was explained accordingly.
  • The figures and legends were changed accordingly.
  • This is a preliminary study, and therefore, the most sensitive enzyme was used to perform docking and MD studies.
  • This is a preliminary study, and therefore, the most potent compound was used to perform MD studies.

Reviewer 3

  • The suggested references were added to the introduction
  • There was a typing mistake and the numbering of compounds 5a-m was corrected
  • All of the other comments regarding experimental and supplementary materials were changed accordingly.

Reviewer 2 Report

The authors present the synthesis and characterization of anilinopyrimidine derivatives as CDK7,8 and 9 inhibitors. In general, the work is good and will be of interest to those working in anticancer drug design. However, there are some points to take into account.

1.- Docking site is not indicated in methodology section. According to the text it can be assumed that the site was that of the ligand in the crystal structure used. If this is true, first, it needs to be indicated in methodology and second, a validation of the docking protocol using this ligand should be included.

2.- Quality and description in figure legends of Figures 3 and 4 need to be improved, some subsections are not easy to understand

3.- In figures 5 and 7 the last 5 ns are missing.

4.- Why do not to perform docking studies in CDK7 and 8 trying to explain the selectivity observed for compounds 5b and 5d?

5.- MD studies with compound 5d must be included because it showed practically the same potency of compound 5b in CDK9.

6.- Discussion of the results is missing, therefore, the relevance of the work is not clear.

Author Response

(The authors gave the same response as above.)

Reviewer 3 Report

This manuscript describes the preparation and anticancer activity of a series of 21 substituted aniline-pyrimidine derivatives. Additionally, obtained compounds were tested as a potential cyclin-dependent kinases inhibitors. It turned out that three of them present very interesting inhibitory potential. Overall, in my opinion the reviewed manuscript presents very interesting biological results, especially in the context of the search for new anticancer agents.
Therefore, I recommend the publication of the reviewed manuscript in Molecules after major revisions according to the following comments.

- in my opinion the introduction should be supplemented with some very relevant references (10.1016/j.bmcl.2017.08.063, 10.1016/j.bmcl.2007.04.021 and F.-W. Sum et al. US Patent 7799915, 2010).

- there is lack of compounds 5i-j, why?

- In the case of known compounds: 3b-d, 3f-h, 5a, 5c, 5g, 5h, 5k and 5o (based on reaxys database), measured melting points should be presented together with literature values and appropriate reference(s).

- Please, reanalyze all MS data of compounds 5a-o. Mentioned MS spectra are obtained at negative mode, however in most cases described molecular peak is assigned as [M++1], what is unusual for this type of ionisation.
For example spectrum of compound 5l (page 37 in supplementary material) was obtained at negative mode, while the molecular peak was described as M+1=328.99 (manuscript, page 24, line 584). At first, there is lack of signal 328.99. Secondly, in fact there is a strong peak 327.05 which correspond to [M-1] peak.
Generally, in the case of negative mode, peak [M-1]  (not M+1) should be observed.

- page 3, line 76-78, figure 2: the –CN group selection should be better justified.

-page 7 line 143: in the case of chlorine (and other halogens) the phrase “a small substituent with inductive effect” will be better than “small electron withdrawing substituent” (in fact Cl is pi-donating substituent, however the inductive effect prevails over the electron donation).

-page 7 line 146: OCH3 is not an bulky group.

-page 21 line 441: check the signal at 7.49 ppm (“ddd” multiplicity cannot appear in this case). Please, reanalyze it.

-page 21 line 460, 13C NMR: the CF3 carbon signal is a quartet not singlet (it is clearly visible on 13C NMR, see page 8 in supplementary material), please correct it.

-page 21 line 466-467, 13C NMR: the OCF3 carbon signal is a quartet not singlet (it is clearly visible on 13C NMR, see page 10 in supplementary material), please correct it.

-page 21 line 470: there are too many protons on the 1H NMR description.

-page 22 line 482, 13C NMR: there is lack of carbon-fluorine coupling. Please, reanalyze 13C NMR spectrum and complete it.

-page 22, point 4.1.3.2. there is lack of M.p. for guanidine 4b.

- page 22, line 515-516, 13C NMR of 5a: this spectrum was over interpreted, what’s more some signals are duplicated. There are only 7 signals clearly visible on 13C NMR spectrum (see page 18 in supplementary material). In my opinion 13C NMR spectrum have to be repeated (use a larger sample or longer acqusition time).

-page 23 line 537, 13C NMR of 5d: omitted two lines at ca.123 and 125 ppm (see page 24 insupplementary material ) are in fact the middle two lines of the quartet of CF3 carbon (the flanked lines of this quartet have very low intensivity). Please, reanalyze 13C NMR spectrum of 5d.

-page 21 line 545, 13C NMR of 5e: the OCF3 carbon signal is a quartet not singlet (it is clearly visible on 13C NMR, see page 26 in supplementary material), please correct it.

-page 23 line 552-553, 13C NMR of 5f: there is lack of carbon-fluorine coupling, please reanalyze carefully 13C NMR spectrum of 5f and complete 13C NMR description.
The same applies compound 5n (page 24, line 597-598).

Comments to the Supplementary materials:

-the copies of the NMR spectra (especially 1H NMR), are low quality. Furthermore, copies of full spectra are not very readable for the potential readers. Thus, additional copies/images of the expanded fragments of 1H NMR spectra should be added  (like for example it was done in the case of 13C NMR of 3b or 1H NMR of 5d).

- Page 2: 13C NMR of 3a: please add new copy of this spectrum (current image was partially “cut”).

- Page 20: 13C NMR of 5b: please add new copy of this spectrum (current image was partially “cut”)

- page 30: there is lack of 1H NMR spectrum of 5g. 13C NMR spectrum is duplicated. Please correct it.

- page 32: there is lack of 13C NMR spectrum of 5h. 1H NMR spectrum is duplicated. Please correct it.

Author Response

(The authors gave the same response as above.)

Round 2

Reviewer 2 Report

In this revised version, some of the concerns form the first manuscript were corrected. However, two of them are still unsolved, to include in methodology the docking site, and the most important, there is not a discussion section about the results, all the manuscript is just a description of the data, therefore, as was mentioned previously, the contribution of the work, according to the results analysis, is missing.

Author Response

Thank you so much for your helpful comments.

The docking site was indicated in molecular docking sections and highlighted in light blue. In addition, the molecular docking results were discussed at the end of section 2.4 and highlighted in light blue.

Reviewer 3 Report

The authors have responded to most of my questions and comments, and made appropriate corrections in the text of the revised manuscript. However, there are still some errors and some comments were not included in the authors’ response.

Therefore, I recommend the publication of the reviewed manuscript in Molecules after minor revisions according to the following comments.

-Page 4, line 72-74, figure 2: the CN group selection should be better justified.

-The 13C NMR spectra of fluorinated compounds are still incorrectly described. The CF3 carbon quartets mentioned in my previous review have to be described as “q” with appropriate C-F coupling constant (as it was done in the case of 5e, page 30, line 529), not by listing of all four lines of quartet. This applies to 13C NMR spectra of compound 3d (page 26 line 448-450), 3e (page 27 line 454-455), 3h (page 27 line 469-470), 5d (page 29 line 521-522), 5f (page 30, line 534-535) and 5l (page 31, line 574).

-Page 22, line 515-516, 13C NMR of 5a: as it was mentioned in my previous review, this spectrum was over interpreted, what’s more some signals are duplicated. There are only 7 signals clearly visible on 13C NMR spectrum. 13C NMR spectrum have to be repeated (use a larger sample or longer acqusition time)

Comments to the Supplementary materials:

-page 6, 1H and 13C NMR of 3c: copy of the 13C NMR spectrum is shown as 1H NMR and vice versa.

-page 8, there is lack of 13C NMR spectrum of 3d (1H NMR is duplicated).

-page 10, 13C NMR of 3e: please add new copy of this spectrum (current image was partially “cut”).

Author Response

Thank you so much for your helpful comments.

All of the suggested modifications were made and highlighted in light blue.